# Evaluation of Spectral Imaging for Early Esophageal Cancer Detection

**DOI:** 10.3390/cancers17122049

**Published:** 2025-06-19

**Authors:** Li-Jen Chang, Chu-Kuang Chou, Arvind Mukundan, Riya Karmakar, Tsung-Hsien Chen, Syna Syna, Chou-Yuan Ko, Hsiang-Chen Wang

**Affiliations:** 1Division of Gastroenterology and Hepatology, Department of Internal Medicine, Ditmanson Medical Foundation Chia-Yi Christian Hospital, Chiayi 60002, Taiwan; cych07235@gmail.com (L.-J.C.); vacinu@gmail.com (C.-K.C.); 2Obesity Center, Ditmanson Medical Foundation Chia-Yi Christian Hospital, Chiayi 60002, Taiwan; 3Department of Medical Quality, Ditmanson Medical Foundation Chia-Yi Christian Hospital, Chiayi 60002, Taiwan; 4Department of Mechanical Engineering, National Chung Cheng University, 168, University Rd., Min Hsiung, Chiayi 62102, Taiwan; d09420003@ccu.edu.tw (A.M.); karmakarriya345@gmail.com (R.K.); 5Department of Internal Medicine, Ditmanson Medical Foundation Chia-Yi Christian Hospital, Chiayi 60002, Taiwan; cych13794@gmail.com; 6Department of Computer Science and Engineering, Chitkara University, Chandigarh-Patiala National Highway NH-64 Village Jansla, Rajpura 140401, Punjab, India; synab6498@gmail.com; 7Department of Gastroenterology, Kaohsiung Armed Forces General Hospital, 2, Zhongzheng 1st Rd., Lingya District, Kaohsiung City 80284, Taiwan; 8Department of Medical Research, Dalin Tzu Chi Hospital, Buddhist Tzu Chi Medical Foundation, No. 2, Minsheng Road, Dalin, Chiayi 62247, Taiwan; 9Department of Technology Development, Hitspectra Intelligent Technology Co., Ltd., Kaohsiung 80661, Taiwan

**Keywords:** esophageal cancer, SAVE, machine learning, artificial intelligence, YOLO, narrow-band Imaging, white-light Imaging

## Abstract

Esophageal cancer (EC) is the eighth most prevalent cancer worldwide and the sixth leading cause of cancer-related mortality. EC must be detected early to prevent patient decline. This study uses SAVE technology for the hyperspectral conversion of an EC image to acquire spectrum information. YOLOv9, YOLOv10, RT-DETR, Roboflow 3.0, and YOLO-NAS models are used to assess EC spectrum data, using deep learning. This study develops a deep learning-based model to identify the stage and anatomical location of malignant esophageal tumors. When specificity and sensitivity were assessed across all the models, SAVE outperformed WLI. In regard to the assessment, SAVE improved the precision and F1 scores for most of the models, which is significant in regard to patient care and therapy. Although Roboflow 3.0 is more sensitive to SCCs, YOLO-NAS operates effectively in all cases. These results show how different machine learning models perform after applying the recommended SAVE imaging approach, under clinically relevant conditions.

## 1. Introduction

Esophageal cancer (EC) represents a major global health burden, ranking as the eighth most prevalent malignancy and the sixth leading contributor to cancer-related mortality worldwide. Epidemiologic analyses reveal striking geographic disparities, with nearly 80% of cases concentrated in socioeconomically disadvantaged regions. A pronounced gender disparity exists, as males constitute approximately 70% of EC patients, exhibiting 2–5 times higher incidence and mortality rates compared to females [1,2]. Population-level survival metrics further underscore systemic healthcare challenges, serving as critical benchmarks for evaluating national cancer control efficacy [3]. According to 2019 estimates from the American Cancer Society, EC claims over 604,100 lives annually [4], emphasizing the urgent need for improved diagnostic strategies. While early detection substantially enhances curative potential [5], current clinical imaging modalities remain insufficient, evidenced by a dismal 5-year relative survival rate of 20%, positioning EC among the malignancies with the poorest prognoses [6].

Dysplasia is the earliest known precursor lesion of the esophageal mucosa and the first step in the progression leading to EC [7]. Dysplasia is a proliferation of disordered cells in the epithelium due to genetic alterations, with a predisposition to invasion and metastasis [8]. The primary histological variants of esophageal cancer are squamous cell carcinoma (SCC) and adenocarcinoma. SCC originates in the squamous epithelium, located in the mid and upper esophagus, and is prevalent in Asia and Africa, whereas ADC arises from the glandular epithelium in the lower esophagus, frequently linked to Barrett’s esophagus, and is more common in Western countries. Their separation is primarily due to distinct cellular backgrounds, varying case frequencies across populations, endoscopic appearances, and specific genetic factors, necessitating tailored screening and treatment strategies for each type. Tobacco use and excessive alcohol consumption are the primary contributors to SCC, and they synergistically increase the susceptibility of squamous mucosa to malignancy. Adenocarcinoma is more prevalent in individuals with obesity and metabolic syndrome, potentially resulting in frequent gastroesophageal reflux disease (GERD) and subsequently leading to Barrett’s esophagus. A diminished consumption of fresh fruits and vegetables, coupled with a heightened intake of nitrosamine-laden foods, may contribute to an elevated risk of esophageal cancer, particularly SCC, in regions where esophageal cancer is prevalent. The individual’s location of residence and financial status influences the likelihood of them having different subtypes: individuals in low-resource areas are more susceptible to SCC due to inadequate nutrition and prevalent alcohol/tobacco consumption, whereas an increasing number of individuals in Western countries are developing adenocarcinoma as obesity and GERD become more prevalent. SCC is the predominant histological variant, constituting 80% of EC cases globally. SCC is defined as a malignant epithelial tumor characterized by squamous differentiation, rather than necessarily originating from the squamous epithelium. Advanced SCC is characterized by a protruding mass or an ulcerated lesion that appears depressed. SCC poses diagnostic challenges due to its frequently nonspecific macroscopic features or color variations; it typically presents as an irregular mucosal surface, enveloped in a thin white coating or exhibiting a reddish tint [9].

Recent advancements in machine learning have significantly propelled EC diagnostic methodologies. Tang et al. pioneered a deep learning framework leveraging esophageal wall thickness measurements from non-contrast chest CT scans, demonstrating sensitivity and accuracy ranges of 75–81% and 75–77%, respectively [10]. While CT-based approaches provide anatomical insights, their clinical utility is constrained by ionizing radiation exposure and suboptimal soft tissue differentiation compared to white-light imaging (WLI). Addressing endoscopic automation challenges, Liu et al. implemented a dual-modal deep learning architecture integrating depth sensing and RGB data, achieving remarkable performance metrics, a 97.54% early EC detection rate and a 74.43% mean Dice coefficient for lesion segmentation [11]. Nakagawa et al. further advanced depth-specific diagnostics through an AI system capable of distinguishing SM1 from SM2/3 submucosal invasions, with 91.0% accuracy, 90.1% sensitivity, and 95.8% specificity [12]. Complementing these efforts, Smith et al. validated a multispectral endoscopic scattering technique, attaining 96% sensitivity and 97% specificity for high-grade dysplasia detection, exceeding American Society for Gastrointestinal Endoscopy (ASGE) benchmarks [13]. Despite these innovations, conventional RGB and multispectral imaging remain fundamentally limited by sparse spectral sampling and discontinuous wavelength coverage at the individual pixel level.

Hyperspectral imaging (HSI), or imaging spectroscopy, captures spatial and spectral data across hundreds of contiguous wavelengths, generating a three-dimensional “hypercube” that maps tissue composition and physiological properties for each pixel [14]. Unlike conventional imaging modalities, HSI provides a dense spectral signature for every spatial point, enabling the detection of subtle mucosal and submucosal abnormalities that are invisible under standard white-light imaging (WLI) [15,16,17,18]. Commercially available HSI systems operate across ultraviolet (200–380 nm), visible (380–780 nm), and near-infrared (780–2500 nm) spectra, offering unparalleled resolution for distinguishing pathological features [19,20]. Narrow-band imaging (NBI), a specialized form of HSI, enhances the vascular and mucosal contrast by selectively illuminating tissues with 415 nm (blue) and 540 nm (green) wavelengths [21]. Blue light penetrates deeper submucosal layers to highlight the subsurface vasculature, while green light reflects superficial capillaries, improving the visualization of early neoplastic changes [22]. For instance, Zhang et al. demonstrated that magnifying endoscopy with NBI (ME-NBI) significantly outperformed WLI in diagnosing early gastric cancer, achieving sensitivity and specificity values of 0.83 and 0.96 compared to 0.48 and 0.67 for WLI [23]. Despite its diagnostic potential, NBI remains underutilized due to hardware limitations in regard to many endoscopy systems. To address this gap, our study integrates HSI with a novel spectral band selection algorithm, the Spectrum-Aided Vision Enhancer (SAVE), to transform conventional WLI endoscopic images into high-fidelity spectral representations. By coupling the SAVE with advanced deep learning architectures (YOLOv9, YOLOv10, YOLO-NAS, RT-DETR, and Roboflow 3.0), we aim to improve the detection of EC stages and anatomical locations. This approach leverages hyperspectral data to train models for identifying dysplasia and squamous cell carcinoma (SCC), overcoming the spectral sparsity of traditional WLI.

## 2. Materials and Methods

### 2.1. Dataset

The dataset for this study includes data collected from Ditmanson Medical Foundation Chia-Yi Christian Hospital. Initially, the dataset comprised 2063 images, categorized into normal, inflammation, dysplasia, and SCC. It includes patient data classed as normal, inflammation, and patients with SCC and dysplasia. A total of 143 patients were enrolled in this study and stratified into three groups based on the histopathological findings: 67 individuals with no evidence of dysplasia, 47 patients with confirmed dysplasia, and 29 patients with biopsy-proven SCC. Among the 143 patients included in the study, 125 were male and 18 were female, with an age distribution ranging from 50 to 75 years old. The patients were selected based on their clinical need for an endoscopic examination, typically due to their symptoms. The majority of the tumors detected during the endoscopy were small, typically less than 1 cm in size, reflecting the goal of early detection. Endoscopy was used as a diagnostic tool, with the primary aim of identifying early-stage esophageal cancer, particularly in patients at higher risk of developing dysplasia or SCC, as early intervention is crucial for improving patient outcomes. The tumor locations and sizes were assessed during the procedure, and the primary focus was on detecting small lesions that could be challenging to identify using conventional WLI. Only images exhibiting clear visualization of the esophageal mucosa and discernible characteristics corresponding to the diagnostic categories were included. The inclusion criteria mandated superior image quality, the absence of motion blur or obstructions, and uniform categorization into one of four specified classes: normal, inflammation, dysplasia, and SCC. Inflammation is defined as benign esophagitis without any dysplastic or neoplastic changes, distinguishing it from malignant conditions. Specifically, the mucosa displayed signs of non-malignant esophagitis, such as erythema, edema, a loss of normal vascular pattern, or superficial erosions, consistent with reflux or infectious esophagitis. Images of inferior quality, characterized by inadequate illumination, substantial artifacts, or deficient clinical annotation, were omitted from the dataset. This stringent selection process guaranteed that the data utilized for model training and evaluation accurately reflected clinically pertinent conditions and reduced potential biases arising from variability in image quality. Prior to training, each image was auto-oriented by stretching, as necessary, and resized to 640 × 640 pixels. To improve model generalization and reduce overfitting, an extensive array of data augmentation techniques was utilized on the training images. Each original image was augmented to generate two supplementary versions, thereby tripling the volume of training data. The augmentations comprised horizontal flipping (50% probability), fixed random rotations of 90°, and random rotations between −10° and +10°, along with shearing transformations within a ±10° range. These augmentations were uniformly implemented across all lesion categories to preserve the class equilibrium (see Appendix A which shows the dataset instances of the classes before and after augmentation). The frequency of each augmentation was randomized for each image to guarantee diversity in the augmented dataset. This method sought to replicate authentic variations in endoscopic imaging, including diverse camera angles and tissue deformations, thus enhancing the resilience of deep learning models to discrepancies observed in clinical practice. The augmentation strategy enhanced the models’ capacity to generalize in regard to novel data during validation and testing by expanding the diversity of the training set. A systematic hyperparameter tuning process was implemented for each deep learning model to enhance model performance and ensure reproducibility. The main hyperparameters that were modified comprised the learning rate, batch size, weight decay, choice of optimizer, and detection confidence threshold. Each model was trained utilizing the Adam optimizer, with the selection corroborated by preliminary experiments assessing the convergence rates and validation loss. The confidence threshold for object detection was adjusted between 0.4 and 0.7 to optimize precision and recall. Early stopping and learning rate scheduling were utilized to mitigate overfitting and guarantee stable convergence. The final hyperparameters for each model were determined based on the model’s performance in regard to the validation set, utilizing the F1 score and mean Average Precision (mAP) as the principal evaluation metrics. All the tuning procedures were uniformly executed across the WLI and SAVE datasets to guarantee an equitable comparison. The overall flowchart of the study is shown in Figure 1.

### 2.2. SAVE

The SAVE conversion approach proposed in regard to this novel algorithm provides an efficient method to transform WLI images that are captured by endoscopes into NBI images. The calibration performed in regard to the series of images used the macbeth color checker (X-Rite classic) and compared the source WLI value with the spectrophotometer data, collected from a spectrometer. Initially, the raw pixel data were linearized and normalized to obtain WLI values in the sRGB color space; these values were then converted into the CIE 1931 XYZ color space via a nonlinear correlation matrix and coefficient. Specifically, the R, G, and B values (0–255) in the sRGB color space were first converted to a smaller range between 0 and 1. The gamma function was then used to convert the sRGB values to linearized RGB values. After the error was corrected, the X, Y, and Z values were updated (XYZ_correct_) and calculated as follows:(1)$$\left[C\right]=\left[{XYZ}_{Spectrum}\right]\times pinv\left(\left[V\right]\right),$$(2)$$\left[{XYZ}_{Correct}\right]=\left[C\right]\times [V].$$

The reflectance spectra from the spectrometer were transformed into the XYZ color space by using the integrals. The correction coefficient matrix C was obtained by performing multiple regression. The reflectance spectrum data (R_spectrum_) were used to obtain the transformation matrix (M) of the colors in regard to the X-Rite board. R_spectrum_ was subjected to PCA to extract six important principal components (PCs) and eigenvectors. These six PCs can account for 99.64% of the information.

The resulting analog spectrum ${\left[S_{\mathrm{Spectrum}}\right]}_{380\sim 780nm}$ can be derived using(3)$$\left[M\right]=\left[Score\right]\times pinv\left(\left[V_{Color}\right]\right)$$(4)$${\left[S_{\mathrm{Spectrum}}\right]}_{380\sim 780nm}=\left[EV\right]\left[M\right][V_{Color}]$$
with an average RMSE of 0.056, validating the high fidelity of the spectral reconstruction. Before the camera was calibrated, the mean chromatic aberration across all 24 color blocks was 10.76. The average chromatic aberration decreased to as low as 0.63 after the camera was calibrated, proving the increased accuracy of the better-vectorized kernels. The color match was further quantified in regard to the LAB color space by using qualitative analysis, which showed that the colors were indeed similar. An average color difference of 0.75 was obtained, which further supports the competence of the given algorithm in regard to the conversion of color from WLI images to hyperspectral images.

The SAVE technique used in this study established a high-level method to calibrate the NBI for the Olympus endoscope on the basis of a HSI conversion algorithm. The simulation process started with a comparison that proved that the images generated by the conversion algorithm resembled realistic NBI images that the Olympus endoscope can produce. Each of the 24 color blocks had its CIEDE 2000 color difference measured and, subsequently, reduced. The average color difference of the 24 color blocks was determined to be negligible (2.79) after the modification. The accuracy of the color match was primarily influenced by three factors. The first factor concerned the light spectrum, which corresponds to the range of wavelengths of light. Then, the color matching function, which is a mathematical function that notifies the amount of light stimulus for a particular color. Finally, the reflection spectrum was the last factor, which contains information about the amount of light reflected by a particular color [24]. The lighting spectra were adjusted, according to the Cauchy–Lorentz distribution, as follows, to minimize the sources of error caused by different lighting spectra of the WLI and NBI, particularly within the 450–540 nm region of high hemoglobin absorbance:(5)$$f\left(x;x_{0},\gamma \right)=\frac{1}{\pi \gamma \left[1+{\left(\frac{x-x_{0}}{\gamma }\right)}^{2}\right]}=\frac{1}{\pi }\left[\frac{\gamma }{{\left(x-x_{0}\right)}^{2}+\gamma^{2}}\right].$$

Thus, the fine tuning of the lighting spectrum was adjusted using the optimization function, namely the dual annealing mechanical method, which is a modification and enhancement of the simulated annealing algorithm that is based on the classical and fast simulated annealing, with local search parameters [25]. The 24 colors had a negligible average standard CIEDE 2000 color difference of 3.06. Incorporating the additional narrow bands as 600, 700, and 780 nm apart from the peak image formation wavelengths of 415 and 540 nm provided a better resemblance of the actual NBI images because small post-processing effects considered in actual captures of the Olympus NBI were recreated. The structural similarity index metric (SSIM), entropy, and peak signal-to-noise ratio (PSNR) [26] were used to test the simulated NBI. The SSIM achieved a value of 94.27%, the entropy difference for the Olympus endoscope was 0.37% on average, and the PSNR of the Olympus images was 27.8819 dB. This software-based method enhances standard WLI images without the need for specialized NBI hardware, providing a practical and accessible tool for improving lesion detection in clinical endoscopy.

### 2.3. ML Algorithms

#### 2.3.1. YOLOv10

YOLOv10 can be considered as the newest iteration of YOLO models, which demonstrates increased speed and significantly improved accuracy [27]. This version employs a better backbone network and supplements features, such as attention mechanisms, to improve feature extractions. The classification loss is the cross-entropy between the predictions (preds) and the ground truth class labels, batch[“cls”]. It can be calculated as follows:(6)$$\mathrm{Loss}=\mathrm{Cross}\;\mathrm{Entropy}\;\left(\mathrm{preds},\;\mathrm{batch}\;\left[”{cls}^{2}\right]\right)$$

The loss combines one-to-many detection losses, one2many, and one-to-one losses, one2one. It aggregates both to calculate the final detection loss with respect to bounding boxes and classifications. It can be calculated as follows:(7)$$\mathrm{Total}\;\mathrm{Loss}={\mathrm{Loss}\;}_{\mathrm{one}2\mathrm{many}}+{\mathrm{Loss}\;}_{\mathrm{one}2\mathrm{one}}$$

This set of equations allows for a conceptual framework for each loss type used in these classes because they apply to bounding box regression, keypoint estimation, segmentation, and classification across various model contexts.

#### 2.3.2. YOLOv9

YOLOv9 brings numerous innovations to the field of real-time object detection, including programmable gradient information (PGI) and a generalized efficient layer aggregation network (GELAN) [28].

PGI speeds up the gradient computation through the use of an auxiliary reversible branch, which improves the training of the models. In addition to the methods of operation of the abovementioned aspects, GELAN increases the parameter usage, coupling it with more computational efficiency, which could be realized after building its concept on CSPNet and the addition of ELAN. The distribution focal loss (DFL) is calculated for all distances between the estimated and target left/right boundaries, as follows:(8)$$\mathrm{DFL}\;\mathrm{Loss}=\frac{\sum (\mathrm{Cross}\;\mathrm{Entropy}(\mathrm{pred}\text{\_}\mathrm{dist},\;\mathrm{target}\text{\_}\mathrm{left})\cdot \;\mathrm{weight}\text{\_}\mathrm{left}\;+\;\mathrm{Cross}\;\mathrm{Entropy}(\mathrm{pred}\text{\_}\mathrm{dist},\mathrm{target}\text{\_}\mathrm{right})\;\cdot \;\mathrm{weight}\text{\_}\mathrm{right})\;}{\;\mathrm{target}\text{\_}\mathrm{scores}\text{\_}\mathrm{sum}\;}$$

The computed loss, also known as the classification loss, uses binary class-entropy, with optional label smoothing. It can be calculated as follows:(9)$$\mathrm{Classification}\;\mathrm{Loss}=\frac{\sum BCE\;(\mathrm{pred}\text{\_}\mathrm{scores},\;\mathrm{target}\text{\_}\mathrm{scores})}{\;\mathrm{target}\text{\_}\mathrm{scores}\text{\_}\mathrm{sum}\;}$$

The bounding box (bbox) loss in regard to the computed loss combines the loU Loss and the DFL, if applied. The two separate BboxLoss calculations for the bbox loss and bbox loss2 are weighted with the target scores, as follows:(10)$$\mathrm{Bounding}\;\mathrm{Box}\;\mathrm{Loss}\;=\frac{\sum \;\left(\mathrm{IoU}\;\mathrm{Loss}\;+\;\mathrm{DFL}\;\mathrm{Lass}\;\left(\mathrm{if}\;\mathrm{enabled}\right)\right)}{\;\mathrm{target}\text{\_}\mathrm{scores}\text{\_}\mathrm{sum}\;}$$

The above set of equations defines a few variants of loss functions that were used across the classes for bounding box regression, classification, and focal losses. Each loss has balancing and scaling parameters. In every case, the losses were predefined such that correct gradient updates were guaranteed by the training process.

#### 2.3.3. YOLO-NAS

YOLO-NAS is an improved version of the YOLO object detection model, utilizing neural architecture search (NAS) for designing architectures that are specialized for certain tasks [29]. The manual design of the network architecture is bypassed, which also makes YOLO-NAS work more efficiently and provides more accurate results. The model’s training involves three main loss components: classification loss (cls), DFL, and localization loss (loc) [30]. The overlap between the ground truth bounding box and the predicted bounding box is measured by PPYoLOELoss/loss_iou. It attempts to maximize this overlap by increasing the precision of the results of the bounding box prediction. The formula for the computation of the PPYoloELoss/loss is as follows:(11)$$Loss={Loss}_{cls}+\lambda_{iou}{Loss}_{iou}+\lambda_{df}{Loss}_{df}.$$

#### 2.3.4. RT-DETR

RT-DETR is an improved object detection model for real-time applications on the basis of the transformer and the detection framework [31]. The global context and additional detection accuracy are considered with the help of a transformer-based encoder–decoder model. The overall loss function of the proposed approach is given by the following formula:(12)$$L=\alpha L_{aux}+\beta L_{o2o}+\gamma L_{o2m},$$
where $L_{aux}$ gives the dense supervision to the encoder, $L_{o20}$ enriches the one-to-one supervision information of the decoder without losing the characteristics of the end-to-end prediction, and $L_{o2m}$ provides one-to-many dense supervision to the decoder. By default, the loss weights α,β, and γ are set to 1.

RT-DETR’s transformer architecture makes the efficient processing of images within the architecture possible, thereby making it suitable for object detection for real-time applications that require accuracy and speed. Compared with the YOLO models, RT-DETR offers several advantages. The architecture of RT-DETR, based on transformers, is superior to that of YOLO models, on the basis of convolutional neural networks, in terms of global context understanding. So, it performs better in regard to movements, numerous objects, or other complex situations. RT-DETR can handle objects of different sizes and shapes better because they operate in parallel, while focusing on a certain area of the image, unlike YOLO models where the size of the objects is read through the boxes corresponding to the grid cells of the image. Even though YOLO models are renowned for their real-time execution, RT-DETR’s comparable speed comes with the capacity for even higher accuracy because of its more developed attention mechanisms and effective utilization of computational resources.

#### 2.3.5. Roboflow 3.0

Roboflow 3.0 is an end-to-end ML platform, especially for computer vision, with features for datasets and model management and deployment [32]. This version adds new features for training process automation, hyperparameter tuning, and the integration of modern models, such as YOLO-NAS, YOLOv10, and RT-DETR. Roboflow 3.0 is characterized by a simple interface that makes it easy to manage data annotations, augmentation, and version control. The angles used to represent the image information are compatible with all the attached object detection models and enable the training and evaluation of models with nonstandard loss functions, suitable for specific cases. The classification, objectness, and localization losses are computed according to specific criteria of the particular model under consideration, to discourage low performance in regard to various activities. Roboflow 3.0 has become a vital tool for creating and implementing highly effective computer vision applications, owing to its successful integration with the most popular ML frameworks and through API support.

## 3. Results

This study examined endoscopic pictures obtained from the KMUH, encompassing patients classified as normal, dysplastic, and diagnosed with SCC. The dataset distribution is as follows: 26.5% were grouped as normal, and 30.1% were diagnosed with SCC. Furthermore, 43.4% of the patients were classified as moderate smokers, 50.6% were smokers, and 6% had never smoked. Alcohol consumption was prevalent among 45.8% of the patients, with 43.4% engaging in moderate drinking, while 10.8% reported being non-drinkers. The prevalence of betel nut consumption in this group was significant, with 65.1% of patients indicating usage, 16.9% having ceased its use, and 18.1% claiming never having used it. The dataset was randomly partitioned into training, validation, and testing subsets, with a 7:2:1 ratio, respectively. This study aimed to assess the performance of the SAVE imaging modality in comparison with the WLI modality to diagnose several esophageal conditions, including dysplasia, SCC, inflammation, and normal tissue. The evaluation was conducted using different object detection algorithms, namely YOLOv9, YOLOv10, RT-DETR, Roboflow 3.0, and YOLO-NAS.

Based on the YOLOv9 model, the SAVE achieved a higher diagnostic accuracy for most conditions than the previous model (see Appendix A for the visualization of training loss and performance metrics (precision, recall, and F1 score) for YOLOv9 models using both WLI and SAVE imaging modalities. The plots illustrate the learning curves during training and validation phases, highlighting differences in convergence behavior and overall model performance between the two imaging approaches). In the case of precision for dysplasia detection, the figure increased from 70.1% with WLI to 81.3% with the SAVE (see Appendix A for the confusion matrices for YOLOv9 model performance based on WLI and SAVE imaging modalities. The matrices display the classification outcomes for each lesion class (normal, inflammation, dysplasia, and SCC), allowing a comparison of true positives, false positives, and false negatives between WLI and SAVE). The detection of SCC also improved. The analysis of numerous conditions by this model showed that it possesses good generality when strengthened with SAVE images. YOLOv9 performed well compared to the other models, and it was beaten slightly in terms of performance by other models like Roboflow 3.0 and YOLO-NAS (see Appendix A for the F1 score confidence curves for the YOLOv9 model using WLI and SAVE imaging modalities. The curves illustrate the relationship between model confidence thresholds and corresponding F1 scores, demonstrating the impact of varying confidence levels on classification performance for each imaging approach). The method was more accurate in terms of precision and recall for SCC and normal tissue.

The results of the YOLOv10 model emphasized the effectiveness of the SAVE, especially in regard to identifying SCC and normal tissue (see Appendix A for the visualization of training loss and performance metrics (precision, recall, and F1 score) for YOLOv10 models using both WLI and SAVE imaging modalities. The plots illustrate the learning curves during training and validation phases, highlighting differences in convergence behavior and overall model performance between the two imaging approaches). Specifically, the precision for SCC detection was enhanced from 86.6% with WLI to 88.7% with the SAVE (see Appendix A for the confusion matrices for YOLOv10 model performance based on WLI and SAVE imaging modalities. The matrices display the classification outcomes for each lesion class (normal, dysplasia, SCC, and inflammation), allowing a comparison of true positives, false positives, and false negatives between WLI and SAVE). Such enhancements assert that the SAVE modality helps in improving the specificity of the model in detecting details, therefore increasing efficiency in regard to detection (see Appendix A for the F1 score confidence curves for the YOLOv10 model using WLI and SAVE imaging modalities. The curves illustrate the relationship between model confidence thresholds and corresponding F1 scores, demonstrating the impact of varying confidence levels on classification performance for each imaging approach). Even though the YOLOv10 model had a similar performance to YOLOv9, some results were slightly better in regard to the YOLOv10 model. However, YOLO-NAS performed better in regard to dysplasia and inflammation detection and Roboflow 3.0 in regard to SCC detection.

On average, the SAVE performed better than WLI in terms of all the results emerging from the RT-DETR model (see Appendix A for the confusion matrices for RT-DETR model performance based on WLI and SAVE imaging modalities. The matrices display the classification outcomes for each lesion class (normal, in-flammation, dysplasia, and SCC), allowing a comparison of true positives, false positives, and false negatives between WLI and SAVE). The precision in regard to the detection of normal tissue increased from 66.09% for WLI to 75.4% with the SAVE, as shown in Table 1 (see Appendix A for the predicted lesion detection results for WLI and SAVE images using the RT-DETR model. The figure showcases the bounding box predictions and class labels generated by the model, illustrating differences in detection performance between the two imaging modalities). Dysplasia was detected poorly by the model. The recall was as low as 17% when using the SAVE. The findings showed that the RT-DETR model is less capable in regard to this particular medical application than the other models. The RT-DETR model had a lower precision and recall than the other models in almost all the conditions tested.

As demonstrated in Table 1, Roboflow 3.0 showed significant performance improvements with the SAVE, especially in the case of SCC detection, where it had a precision of 95.45%. The highest accuracy was obtained for this model and the F1 score was 90.32%. In regard to the detection of dysplasia, all the measures improved from the baseline values, with the recall increasing from 51.23% with WLI to 53.1% with the SAVE. Thus, Roboflow 3.0 is one of the best models in regard to this use case.

The results revealed that the SAVE constantly exhibited better performance with the YOLO-NAS model across all the conditions concerning dysplasia and inflammation, as shown in Figure 2. The specific precision for dysplasia detection increased from 75.13% with WLI to 79.81% with the SAVE. The best results were observed for the identification of inflammation and dysplasia, although the overall performance differences between the YOLO-NAS and Roboflow 3.0 models were similar to those of the other models. The relatively reduced SCC detection efficiency indicates that more fine tuning is needed for this particular condition.

Comparing the specificity and sensitivity across all the models revealed that the SAVE modality was more accurate than WLI. The evaluation showed that the SAVE enhanced the precision and F1 scores for most models, which are valued factors in terms of patient care and treatment. Of all the models, Roboflow 3.0 and YOLO-NAS were the most efficient, with Roboflow 3.0 being more efficient in regard to SCC detection, and YOLO-NAS showing balanced performance across all the conditions. These results demonstrated how well the suggested SAVE imaging modality performs when used in clinically relevant scenarios to enhance the performance of multiple ML models.

The proposed approach in regard to the YOLOv9 model was reported to be more robust and showed promising results, especially in terms of SCC detection, where it achieved high overall accuracies and F1 scores related to the WLI and SAVE modalities, as shown in Table 2. In contrast to the results obtained for other classes, the dysplasia detection rates were lower, and the recall values indicated a problem identifying this class of data. The performance in regard to inflammation detection was satisfactory, with moderate precision and recall for both modalities. Thus, the YOLOv9 model appears to be reliable, but is not as effective at detecting less apparent classes, such as dysplasia.

Some of the achievements when comparing the YOLOv10 and YOLOv9 models include the following: In regard to the detection of SCC, the precision increased, which contributed to improving the performance of the YOLOv10 model. However, dysplasia detection remained an issue, with relatively low recall values. This finding suggests that even though the YOLOv10 model is capable of identifying unambiguous and well-defined cancer-related diseases, such as SCC, the network struggles to identify early stages of cancer diseases like dysplasia. The detection of inflammation improved, but still produced variable results between the WLI and SAVE modalities.

Like most models, RT-DETR had inconsistent performance. As for the SAVE modality, the precision percentage was 100%. However, dysplasia distinction, especially in WLI, was poor, with low recall and F1 scores. This finding implies that the RT-DETR model is implicitly tuned toward detecting significantly advanced carcinoma types, while overlooking early-stage lesions that are similar to dysplasia. In conclusion, RT-DETR is particularly strong in regard to SCC detection, but fails to exhibit the required level of robustness for the proper detection of all categories of data.

In regard to Roboflow 3.0, the usefulness of the SAVE was evident, with improved performance in regard to most categories, especially dysplasia and SCC. The identification of dysplasia, which was usually difficult across all the models, was more precise and had better recall rates when using the SAVE, indicating that this mode of imaging provides more elaborate and useful information in the early stages of the disease. The detection of SCC achieved its best performance in regard to the SAVE, thus strengthening the modality’s improved efficiency in determining severe and clearly defined cancerous conditions.

The YOLO-NAS model showed that the pattern persisted, with the SAVE exceeding WLI in regard to each of the three evaluation metrics. The normal, dysplasia, and SCC classes demonstrated significantly improved recall and precision with the use of the SAVE. The algorithm’s ability to detect dysplasia, which is important in early-stage EC, was significantly better with the SAVE, achieving better recall and F1 scores than WLI. Figure 3 shows that SCC detection was very accurate when operated under SAVE conditions, and the evaluation metrics were generally high. This finding showed that YOLO-NAS has extensive detection functionality when employed with the SAVE modality.

These results provide evidence that the use of the SAVE modality improves the detection capabilities of all the algorithms over WLI, as shown in Figure 4. This improvement is most apparent in regard to certain classes, such as dysplasia and SCC, in which early-stage cancer and well-defined conditions are better served by the enhanced spectral detail of the SAVE. The overall accuracy, sensitivity, and specificity analyses of the SAVE in regard to most classes of data highlight its advantage over the other modalities in enhancing diagnostic accuracy and averting more diverse types of EC conditions.

Confidence intervals (CIs) at 95% were computed for the performance scores presented in Table 2. In regard to SCC detection utilizing YOLOv9, the WLI F1 score was 84.3%, with a CI of [71.7%, 96.9%], whereas the SAVE F1 score was 90.4%, with a CI of [80.2%, 100%], and *p* = 0.03. In regard to the cases of dysplasia, the F1 score increased from 60.3% (CI: [51.5%, 69.1%]) with WLI to 65.5% (CI: [57.0%, 73.8%]) with the SAVE (*p* = 0.04). Inflammation and normal tissue exhibited improvements in regard to the CI; however, the difference for inflammation was not statistically significant (*p* = 0.12), likely attributable to the limited sample size (*n* = 16). The enhancements in the precision, recall, and F1 score for the SAVE relative to WLI were statistically significant for the majority of classes of data, affirming the robustness of the SAVE across various lesion types and models. This study was conducted using a static dataset consisting of pre-collected and annotated endoscopic images. Real-time detection capabilities were not developed as part of the model evaluation process. However, an application named the Esophageal Cancer Detection Application (EC Detector), designed using the YOLOv9 model, for cancer detection from esophageal images, was developed with the help of PyQt5. The application allows for increased simplicity of working with it, while providing tools for image uploading and analysis. Appendix A reveals the first interactive screen of the tool, and Appendix A illustrates the manner in which images are selected. After the image is chosen, the model highlights areas of possible cancerous tissues, as illustrated in the picture below, by using bounding boxes, in regard to three types of classes, as shown in Appendix A. This application helps with early diagnosis, hence enhancing clinical results. This tool is meant for future modifications, such as the possibility of identifying other cancers or incorporating other ML algorithms. The current SAVE image conversion occurs offline; however, the developed software application facilitates efficient processing and may be modified for real-time applications in subsequent iterations (see Appendix A for the Screenshots of the Windows-based CAD application used for esophageal cancer detection). In contrast to NBI, which necessitates specialized endoscopic equipment, the SAVE improves standard WLI images via a software-based method, providing greater accessibility and the ability to retrospectively apply it to previously acquired images. Furthermore, the SAVE facilitates adaptable spectral feature extraction and deep learning optimization, offering a customizable diagnostic instrument that transcends the rigid parameters of conventional NBI systems. In subsequent endeavors, we intend to incorporate the SAVE imaging and deep learning detection models into a real-time clinical workflow. This will entail enhancing the model inference speed, reducing latency, and verifying performance during live endoscopic procedures. Real-time deployment may substantially improve the early detection of esophageal cancer during routine endoscopy, thereby enhancing decision-making and patient outcomes. The findings unequivocally demonstrated that the SAVE improved the performance of all the evaluated models. The SCC F1 score increased from 84.3% to 90.4% in the YOLOv9 model and from 87.3% to 90.3% in the Roboflow 3.0 model. In the case of dysplasia, YOLOv9 showed an enhancement in precision from 72.4% (WLI) to 76.5% (SAVE), and YOLO-NAS advanced from 75.1% to 79.8%. Roboflow 3.0 had the highest sensitivity (85.7%) and F1 score (90.3%) for SCC detection, using SAVE imaging. YOLO-NAS had the most equitable performance across all lesion categories, rendering it a suitable option for general clinical applications. Of the evaluated models, Roboflow 3.0 demonstrated enhanced sensitivity to SCC lesions, indicating its potential efficacy in high-risk cancer detection endeavors. YOLOv10 and RT-DETR also displayed advantages as a result of the SAVE upgrade, albeit with marginally reduced performance consistency across various lesion types. In contrast, YOLO-NAS exhibited consistent and precise outcomes for normal, dysplastic, and hemorrhagic tissues, providing a versatile solution for thorough esophageal lesion detection. Although the SAVE typically improved the performance of most models, some variability was noted, especially in regard to dysplasia detection. The recall for dysplasia marginally diminished in the RT-DETR model, presumably due to architectural sensitivity or class imbalance. These findings underscore that although the SAVE provides significant overall advantages, variations in model-specific performance necessitate additional examination and refinement. The amalgamation of the SAVE with deep learning algorithms presents a promising non-invasive instrument for the early diagnosis of esophageal cancer. The significant enhancement in the F1 scores and sensitivity demonstrated by the SAVE reflects its ability to identify early-stage or subtle lesions that may be overlooked with WLI. This could assist endoscopists in regard to the early identification of at-risk patients, optimizing intervention timing, and enhancing prognosis in a clinical context.

## 4. Discussion

This study seeks to assess the effectiveness of the SAVE system in enhancing the detection of esophageal cancer lesions, specifically dysplasia and SCC, in comparison to traditional WLI. We evaluated the efficacy of five deep learning models (YOLOv9, YOLOv10, YOLO-NAS, RT-DETR, and Roboflow 3.0) in detecting esophageal lesions through HSI. The amalgamation of the SAVE system with deep learning models markedly improves detection precision, especially for SCC and dysplasia, in contrast to conventional WLI. Of the models assessed, Roboflow 3.0 exhibited superior performance, enhancing its F1 scores and sensitivity. This study illustrates the clinical efficacy of the SAVE system as a dependable instrument for the early identification and diagnosis of esophageal cancer. The primary clinical application of this study is the improvement of EC detection during endoscopic procedures. Although endoscopists will maintain a crucial role in conducting endoscopies, the incorporation of the SAVE system with deep learning models provides substantial assistance in enhancing diagnostic precision. The SAVE system enhances the visual capabilities of conventional WLI, facilitating the detection of subtle lesions, especially early-stage esophageal cancer, dysplasia, and SCC, that might otherwise be overlooked. This technology does not supplant the endoscopist’s role, but rather functions as a significant diagnostic tool, offering real-time, AI-enhanced support in regard to the identification and characterization of lesions. The implementation of the SAVE system may result in enhanced diagnostic accuracy and timeliness, facilitating earlier interventions and improving patient outcomes, especially among high-risk groups. Presently, there exists a paucity of technology capable of directly transforming conventional WLI into HSI for comparative analysis. Although HSI has been investigated in numerous medical applications, its integration with WLI for real-time detection and comparison is still a nascent field of research. Consequently, there is a dearth of research that directly contrasts WLI images with HSI in the realm of EC detection. This study seeks to bridge the existing gap by assessing the efficacy of the SAVE system in transforming WLI images into HSI and improving the identification of esophageal lesions. This innovation offers a new method for enhancing diagnostic accuracy; however, additional progress in HSI conversion technology is required to broaden its applicability in clinical practice.

A notable limitation is that the data were collected solely from one hospital. This limitation may engender bias in regard to the institutions that possess the data and the patients from whom the data were obtained. The validity of the findings is limited, making the study’s conclusions not yet ready for general application. To improve the generalizability of the findings and the proposed model in future research, data must be collected from multiple locations across diverse countries and ethnicities [33]. A subsequent limitation stems from the preprocessing phase, which entails resizing original images of disparate dimensions to a standardized size of 640 pixels per side. This preprocessing method may have neglected essential image data that could have impacted the models’ efficacy, although it was advantageous in optimizing computational resources. Therefore, subsequent research should focus on enhancing the image resolution during preprocessing or employing adaptive resolution algorithms to retain critical data and optimize computational efficiency. Another concern relates to the computational requirements associated with HSI and ML techniques. The intricate calculations involved pose a challenge due to their demand for substantial time and resources. Therefore, future research efforts should investigate the application of enhanced algorithms and the deployment of advanced computing technologies, including graphics processing units (GPUs) and tensor processing units (TPUs). Real-time detection within microseconds requires improvements in both software and hardware, along with the incorporation of specialized accelerators to expedite specific operations, such as GPUs, field-programmable gate arrays, application-specific integrated circuits, and TPUs [34]. Conversely, ensemble learning amalgamates multiple models to enhance precision and bolster the system’s dependability for clinical use [35]. These methods can improve the technology’s usability, greatly assisting healthcare providers, especially endoscopists, and enhancing the reliability of the results they produce. The SAVE methodology utilized in EC could be easily adapted to various cancer types and possibly other medical conditions, where imaging serves as the most effective method for early diagnosis. Incorporating additional cancer types may illustrate the system’s adaptability. The potential of these advancements may significantly improve patients’ quality of life through early detection and timely treatment of conditions. Improvements in medical imaging methodologies, guided by the findings in this study and similar research, may substantially augment diagnostic capabilities and assess patient treatment results. Therefore, future research should rectify the limitations of this study and investigate novel applications of HSI and ML in medical diagnosis. Consequently, improving the technology to include essential advancements, expanding the study population, and exploring diverse conditions to which this device may be relevant could aid in the creation of a dependable diagnostic instrument.

## 5. Conclusions

The findings indicate that the SAVE demonstrates superior accuracy in detecting dysplasia, SCC, and esophageal cancer compared to WLI. Integrating the SAVE with advanced object detection capabilities, including YOLOv9, YOLOv10, RT-DETR, and Roboflow 3.0, resulted in enhancements in diagnostic accuracy, precision, recall, and F1 scores across the different models. Among all the models, Roboflow 3.0 exhibited the best performance. It also establishes a framework for the advantages of the SAVE system, as it can yield significantly more precise and dependable diagnoses, in addition to the essential knowledge required in clinical practice, wherein early disease detection positively impacts patient prognosis, especially in the context of SCC and normal tissue, wherein the SAVE outperformed WLI. Implementing the SAVE system, particularly within cohorts employing effective machine learning techniques, may enhance the probability of early detection of esophageal diseases. Subsequent research should enhance these models, gather data from a more extensive cohort of patients, and explore the application of the SAVE method for the identification of more malignancies. These advancements may improve cancer detection, augment patient treatment, and, hence, elevate the five-year survival rates. The integration of SAVE with deep learning algorithms significantly improves the detection of esophageal cancer lesions, particularly squamous cell carcinoma and dysplasia, compared to conventional white-light imaging. This emphasizes the SAVE’s potential as a crucial clinical tool for the early detection and diagnosis of cancer. In conclusion, our research illustrates that utilizing the SAVE in regard to conventional endoscopic images and integrating the hyperspectral results into contemporary object-detection networks markedly improves the early identification of esophageal dysplasia and SCC. This signifies a significant progression beyond conventional WLI-based AI models and establishes a viable, scalable framework for forthcoming prospective trials and ultimate clinical implementation.

## Figures and Tables

**Figure 1 cancers-17-02049-f001:**
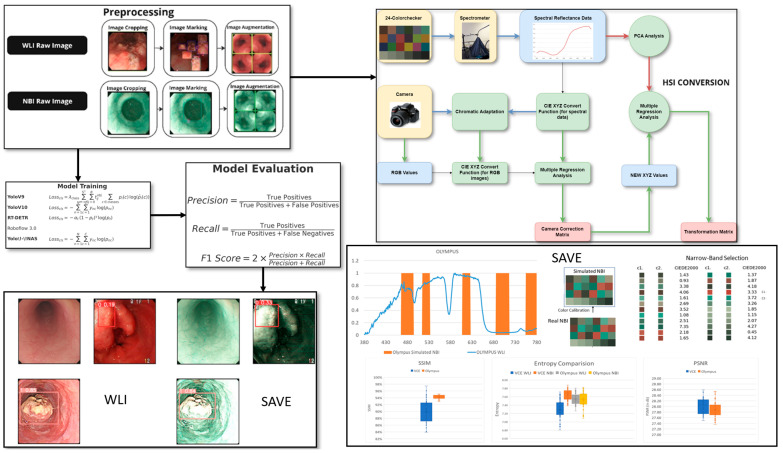
Workflow for the study including data preprocessing and augmentation, application of SAVE for hyperspectral enhancement, and training of deep learning models (YOLOv9, YOLOv10, YOLO-NAS, RT-DETR, Roboflow 3.0) for lesion classification and the final performance of the model was evaluated using precision, recall, sensitivity, and F1 score for both WLI and SAVE conditions.

**Figure 2 cancers-17-02049-f002:**
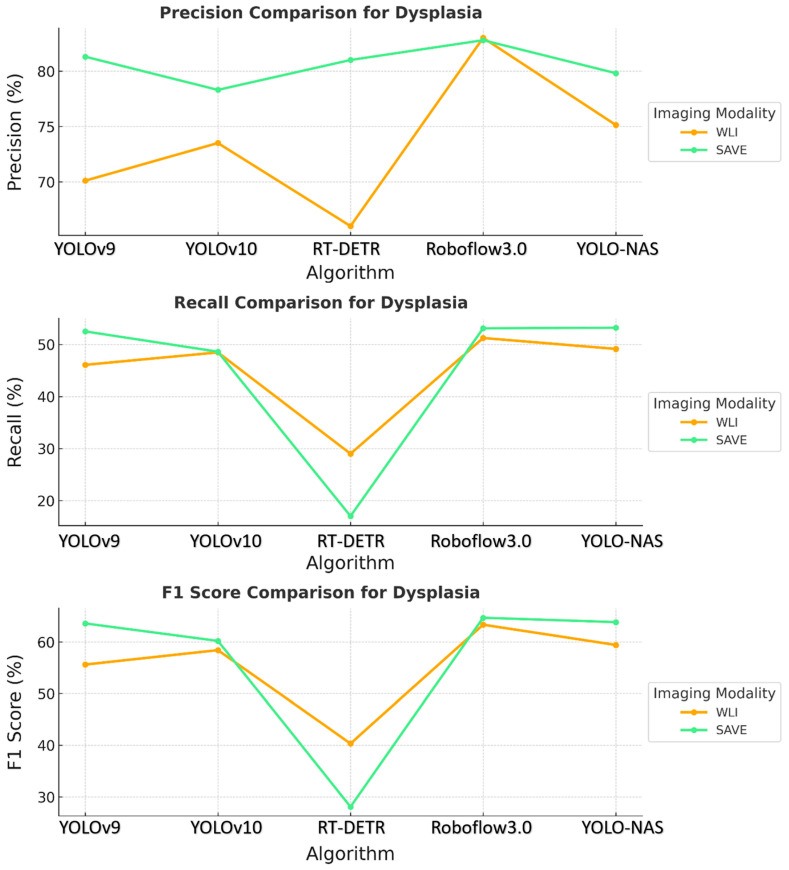
Comparative analysis of precision, recall, and F1 score for the detection of dysplasia using different object detection algorithms (YOLOv9, YOLOv10, RT-DETR, Roboflow 3.0, and YOLO-NAS) across two imaging modalities, namely WLI and SAVE during validation.

**Figure 3 cancers-17-02049-f003:**
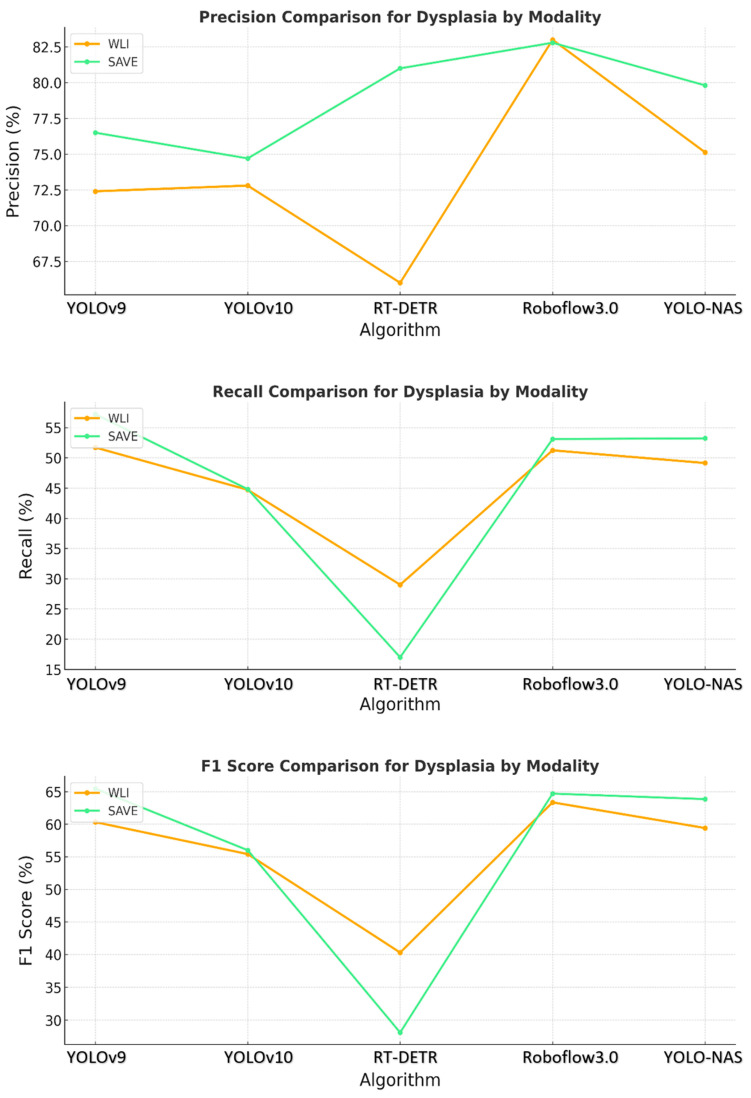
Comparative analysis of precision, recall, and F1 score for the detection of dysplasia using different object detection algorithms (YOLOv9, YOLOv10, RT-DETR, Roboflow 3.0, and YOLO-NAS) across two imaging modalities, namely WLI and SAVE during the testing.

**Figure 4 cancers-17-02049-f004:**
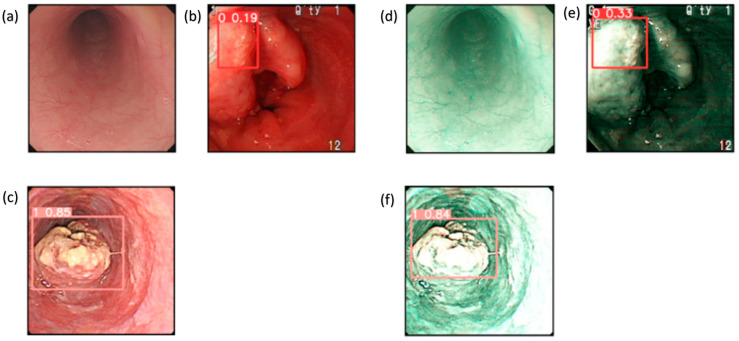
Detection results for WLI and SAVE: (**a**–**c**) show the detection results for the WLI images, which contain normal, dysplasia, and SCC classes, while (**d**–**f**) show the detection results for the SAVE images, which contain normal, dysplasia, and SCC classes.

**Table 1 cancers-17-02049-t001:** Validation results.

Algorithm	Imaging Modality	Class	Precision in %	Recall in %	F1 in %
YOLOv9	WLI	Normal	74.9	72.47	73.7
		Inflammation	86.5	75.4	80.5
		Dysplasia	72.4	51.7	60.32
		SCC	88.6	80.4	84.3
	SAVE	Normal	81.3	71.8	76.29
		Inflammation	71.1	62.8	66.69
		Dysplasia	76.5	57.2	65.45
		SCC	88.7	92.2	90.41
YOLOv10	WLI	Normal	76.5	65.3	70.2
		Inflammation	85.7	56.5	68.1
		Dysplasia	72.8	44.7	55.4
		SCC	94.9	78.3	85.8
	SAVE	Normal	77	73.8	75.3
		Inflammation	65.6	56.8	60.9
		Dysplasia	74.7	44.8	56
		SCC	83.9	81.8	82.8
RT-DETR	WLI	Normal	77.125	60	66.09
		Inflammation	57	66	61.17
		Dysplasia	66	29	40.29
		SCC	75	68	71.32
	SAVE	Normal	75.4	69.97	72.59
		Inflammation	54	38	44.6
		Dysplasia	81	17	28.1
		SCC	100	73	84.39
Roboflow 3.0	WLI	Normal	77.63	70.27	73.63
		Inflammation	68.62	56.45	61.94
		Dysplasia	83	51.23	63.35
		SCC	100	77.5	87.32
	SAVE	Normal	83.51	67.91	74.52
		Inflammation	72.98	62.3	67.22
		Dysplasia	82.79	53.1	64.7
		SCC	95.45	85.71	90.32
YOLO-NAS	WLI	Normal	81.79	62.38	69.95
		Inflammation	68.64	60.99	64.59
		Dysplasia	75.13	49.13	59.41
		SCC	97.82	88.2	92.78
	SAVE	Normal	80.47	67.64	73.36
		Inflammation	73.33	52.38	61.11
		Dysplasia	79.81	53.21	63.85
		SCC	89.65	65	75.36

**Table 2 cancers-17-02049-t002:** Performance metrics for all models under WLI and SAVE imaging conditions. This table summarizes the test results of five deep learning models (YOLOv9, YOLOv10, YOLO-NAS, RT-DETR, and Roboflow 3.0) for esophageal lesion detection, using both WLI and SAVE modalities.

Algorithm	Imaging Modality	Class	Precision in %	Recall in %	F1 in %
YOLOv9	WLI	Normal	74.9	72.47	73.7
		Inflammation	86.5	75.4	80.5
		Dysplasia	72.4	51.7	60.32
		SCC	88.6	80.4	84.3
	SAVE	Normal	81.3	71.8	76.29
		Inflammation	71.1	62.8	66.69
		Dysplasia	76.5	57.2	65.45
		SCC	88.7	92.2	90.41
YOLOv10	WLI	Normal	76.5	65.3	70.2
		Inflammation	85.7	56.5	68.1
		Dysplasia	72.8	44.7	55.4
		SCC	94.9	78.3	85.8
	SAVE	Normal	77	73.8	75.3
		Inflammation	65.6	56.8	60.9
		Dysplasia	74.7	44.8	56
		SCC	83.9	81.8	82.8
RT-DETR	WLI	Normal	77.125	60	66.09
		Inflammation	57	66	61.17
		Dysplasia	66	29	40.29
		SCC	75	68	71.32
	SAVE	Normal	75.4	69.97	72.59
		Inflammation	54	38	44.6
		Dysplasia	81	17	28.1
		SCC	100	73	84.39
Roboflow 3.0	WLI	Normal	77.63	70.27	73.63
		Inflammation	68.62	56.45	61.94
		Dysplasia	83	51.23	63.35
		SCC	100	77.5	87.32
	SAVE	Normal	83.51	67.91	74.52
		Inflammation	72.98	62.3	67.22
		Dysplasia	82.79	53.1	64.7
		SCC	95.45	85.71	90.32
YOLO-NAS	WLI	Normal	81.79	62.38	69.95
		Inflammation	68.64	60.99	64.59
		Dysplasia	75.13	49.13	59.41
		SCC	97.82	88.2	92.78
	SAVE	Normal	80.47	67.64	73.36
		Inflammation	73.33	52.38	61.11
		Dysplasia	79.81	53.21	63.85
		SCC	89.65	65	75.36

## Data Availability

The data presented in this study are available in this article and upon request to the corresponding author.

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
