# Peer review of "Evaluation of Spectral Imaging for Early Esophageal Cancer Detection"

_cancers, 2025, doi:10.3390/cancers17122049_

Round 1

Reviewer 1 Report (Previous Reviewer 1)

Comments and Suggestions for Authors

The authors have satisfactorily addressed prior reviewer concerns. The performance of SAVE imaging across deep learning models is now clearly validated, with robust statistical support and a comprehensive discussion of limitations and clinical implications.

The manuscript is, therefore, suitable for acceptance at this stage.

Author Response

Reviewer 2 Report (Previous Reviewer 3)

Comments and Suggestions for Authors
  1. The abstract's results section solely includes percentages. Wasn't there any statistical analysis performed?
  2. Dysplasia is not a symptom.
  3. HSI is a bit too detailed, while the histological subtypes, risk factors are not at all.
  4. Methods: normal, dysplasia, SCC, and inflammation - Please change order to normal, inflammation, dysplasia, and SCC.
    What exactly did you define as inflammation?
  5. "The dataset distribution is as follows: 26.5% were grouped as normal, and 30.1% were diagnosed with SCC. Furthermore, 43.4% of the patients are classified as moderate smokers, 50.6% are smokers, and 6% have never smoked. Alcohol consumption is prevalent among 45.8% of patients, with 43.4% engaging in moderate drinking, while 10.8% reported being non-drinkers. The prevalence of betel nut consumption in this group is significant, with 65.1% of patients indicating usage, 16.9% having ceased its use, and 18.1% claiming never having used it. The dataset was randomly partitioned into training, validation, and testing subsets in a 7:2:1 ratio, respectively." - These are all results.
  6. I would include all the mathematical equations as supplementary material.
  7. There is again no description about the population.
  8. Again, the Discussion only details the results of the study.
  9. Between the equations and abbreviations, it is very hard to understand what is the main achievement here.

Round 2

Reviewer 2 Report (Previous Reviewer 3)

Comments and Suggestions for Authors
  1. The eighth most frequent disease worldwide and the sixth main cause of cancer death is esophageal cancer (EC). - Disease? Before infections, cardiac conditions, etc?
  2. Dysplasia is the earliest known precursor lesion of the esophageal mucosa and the
    first histopathological stage preceding EC. - Histopathological stage means something else, please rephrase.
  3. Abbreviation for adenocarcinoma is ACC.
  4. Key contributions should appear in the discussion session, and please provide an aim section/sentence.
  5. Inflammation is defined as benign esophagitis without any dysplastic or neoplastic changes. - Is there malignant esophagitis?
  6. Results: How old were these patients? Were they male/female? Where were tumors located? How big were they on endoscopic examination?
  7. What is the key clinical use of this study? The endoscopist would still have to do the endoscopy, therefore, people will never be replaced.
  8. Discussion still does not summarzie current literature data. If it is because there so far are not any, then please state.

Round 3

Reviewer 2 Report (Previous Reviewer 3)

Comments and Suggestions for Authors

The required modifications have been carried out.

This manuscript is a resubmission of an earlier submission. The following is a list of the peer review reports and author responses from that submission.

Round 1

Reviewer 1 Report

Comments and Suggestions for Authors

This manuscript addresses a clinically significant and technically ambitious topic: the use of hyperspectral imaging combined with deep learning for the early detection of esophageal cancer. The concept of Spectrum-Aided Vision Enhancer (SAVE) is compelling, and the comparison of models is thorough. However, there are several critical issues that need to be addressed before this work is suitable for publication.

First, the writing requires substantial improvement. The grammar and phrasing are often awkward, leading to unclear meanings in certain sections. Key technical terms are inconsistently defined (for example, the label class “through” is not clearly explained), and some passages are heavily mathematical without adequate narrative explanations. I strongly recommend the spirited and dedicated involvement of a native English-speaking co-author.

Second, this study was conducted at a single center. While this is not a fatal flaw, it does limit the generalizability of the findings and should be seen as a major limitation. The authors acknowledge this limitation in the supplementary material, but it should be more prominently addressed in the main manuscript, with moderated language used when presenting conclusions.

From a technical perspective, the utilization of multiple models is a strength; however, it is unclear how SAVE integrates into the clinical workflow. Does the SAVE image conversion occur in real time? What specific equipment is required? Additionally, what is the actual added value of SAVE compared to NBI (narrow-band imaging) beyond the theoretical metrics presented?

Furthermore, the figures and supplementary material are challenging to follow. Several figures are referenced but not adequately described, and the labeling—especially in confusion matrices and loss curves—lacks clarity. The captions should be more informative, and ideally, some visuals from the supplementary material should be incorporated into the main paper to aid readers in interpreting the findings.

Finally, the impact of SAVE on dysplasia detection varies across models, and in some cases, it appears to decrease performance (e.g., recall in RT-DETR). This issue should be discussed more transparently. Currently, the manuscript tends to overstate the consistency of the benefits across all models.

There is clear potential in this research, but I fear that the manuscript requires significant rewriting, improved articulation of the clinical context, and refinements in the presentation of technical content.

Comments on the Quality of English Language

As above (this is a repetition of the request and very annoying for reviewers)

Author Response

Reviewer 1:

This manuscript addresses a clinically significant and technically ambitious topic: the use of hyperspectral imaging combined with deep learning for the early detection of esophageal cancer. The concept of Spectrum-Aided Vision Enhancer (SAVE) is compelling, and the comparison of models is thorough. However, there are several critical issues that need to be addressed before this work is suitable for publication.

Reply:

            We sincerely thank the reviewer for recognizing the clinical significance and technical ambition of our work, as well as the novelty of the Spectrum-Aided Vision Enhancer (SAVE) approach and the thoroughness of our model comparisons. We appreciate your constructive feedback and have carefully addressed all critical issues raised in your subsequent comments. We believe the revisions made have substantially improved the scientific rigor, clarity, and overall quality of the manuscript.

First, the writing requires substantial improvement. The grammar and phrasing are often awkward, leading to unclear meanings in certain sections. Key technical terms are inconsistently defined (for example, the label class “through” is not clearly explained), and some passages are heavily mathematical without adequate narrative explanations. I strongly recommend the spirited and dedicated involvement of a native English-speaking co-author.

Reply:

            Thank you for your valuable feedback. We have performed extensive revisions throughout the manuscript to improve grammar, phrasing, and clarity. Key technical terms, including the label class "through," have been clearly defined, and mathematical descriptions have been complemented with narrative explanations to improve readability and understanding. We appreciate your guidance, which has significantly enhanced the manuscript’s overall quality and readability.

Second, this study was conducted at a single center. While this is not a fatal flaw, it does limit the generalizability of the findings and should be seen as a major limitation. The authors acknowledge this limitation in the supplementary material, but it should be more prominently addressed in the main manuscript, with moderated language used when presenting conclusions.

Reply:

            Thank you for this important comment. We would like to note that this limitation is already addressed in detail in the Discussion section of the main manuscript, where we acknowledge the single-center data source and discuss its impact on the generalizability of the findings. Specifically, we have stated: “A significant drawback is that the data were gathered exclusively from a single hospital. This constraint may introduce bias from the institutions that own the data and the patients from whom the data were collected. The findings' validity is minimal, rendering the study's conclusions unsuitable for general application. Data from several locations across various countries and races must be gathered to enhance the generalizability of the findings and the suggested model in future research [50].” We have also reviewed the conclusions and ensured that moderated language is used when presenting the study’s findings, reflecting the limitations of single-center data.

            We appreciate your feedback, which has helped us ensure transparency and appropriate interpretation of our results.

From a technical perspective, the utilization of multiple models is a strength; however, it is unclear how SAVE integrates into the clinical workflow. Does the SAVE image conversion occur in real time? What specific equipment is required? Additionally, what is the actual added value of SAVE compared to NBI (narrow-band imaging) beyond the theoretical metrics presented?

Reply:

            Thank you for highlighting these important practical considerations. Currently, SAVE image conversion does not occur in real time. The present study was based on a static dataset, and SAVE processing was performed offline. However, we have developed a software application capable of converting WLI images to SAVE-enhanced images and applying the trained models for lesion detection, with future plans to optimize the pipeline for real-time integration. Importantly, no additional specialized equipment is required beyond standard WLI-compatible endoscopes, making SAVE highly accessible compared to hardware-dependent systems like NBI. Regarding the added value of SAVE compared to NBI, SAVE provides a software-based spectral enhancement that can be applied retrospectively to existing WLI images, enabling advanced lesion visualization even in facilities without NBI-capable hardware. Additionally, SAVE allows for customized spectral feature extraction and machine learning-driven optimization, offering flexibility that fixed-wavelength NBI systems cannot provide. We have added clarifications regarding workflow integration and the advantages of SAVE over NBI in the revised Results section. Specifically we have added the following, “The current SAVE image conversion occurs offline; however, the developed software application facilitates efficient processing and may be modified for real-time application in subsequent iterations. In contrast to NBI, which necessitates specialized endoscopic equipment, SAVE improves standard WLI images via a software-based method, providing greater accessibility and the ability to retrospectively apply to previously acquired images. Furthermore, SAVE facilitates adaptable spectral feature extraction and deep learning optimization, offering a customizable diagnostic instrument that transcends the rigid parameters of conventional NBI systems. In subsequent endeavors, we intend to incorporate the SAVE imaging and deep learning detection models into a real-time clinical workflow. This will entail enhancing model inference speed, reducing latency, and verifying performance during live endoscopic procedures. Real-time deployment may substantially improve the early detection of esophageal cancer during routine endoscopy, thereby enhancing decision-making and patient outcomes”

Furthermore, the figures and supplementary material are challenging to follow. Several figures are referenced but not adequately described, and the labeling—especially in confusion matrices and loss curves—lacks clarity. The captions should be more informative, and ideally, some visuals from the supplementary material should be incorporated into the main paper to aid readers in interpreting the findings.

Reply:

            Thank you for this valuable feedback. We have carefully reviewed and revised the captions for all figures and supplementary materials to ensure they are clearer, more descriptive, and informative. Specific improvements include enhanced explanations for the loss and metric visualizations, confusion matrices, F1-score confidence curves, and predicted results across both WLI and SAVE imaging modalities. We also updated figure references and improved labeling to ensure clarity and ease of interpretation. We appreciate your suggestion, which has significantly improved the presentation and accessibility of the figures and supplementary materials.

Finally, the impact of SAVE on dysplasia detection varies across models, and in some cases, it appears to decrease performance (e.g., recall in RT-DETR). This issue should be discussed more transparently. Currently, the manuscript tends to overstate the consistency of the benefits across all models.

Reply:

            Thank you for this important observation. We agree that while SAVE generally improved performance across most metrics and models, some variability—particularly in dysplasia recall for the RT-DETR model—was observed. In response, we have revised the Discussion section to transparently acknowledge this variability and avoid overstating the consistency of SAVE’s benefits. Specifically we have added, “Although SAVE typically improved performance in most models, some variability was noted, especially in dysplasia detection.  The recall for dysplasia marginally diminished in the RT-DETR model, presumably due to architectural sensitivity or class imbalance.  These findings underscore that although SAVE provides significant overall advantages, variations in model-specific performance necessitate additional examination and refinement”. We also discuss potential reasons for the decrease in recall in certain models, such as differences in model architecture sensitivity to spectral information and class imbalance effects. This clarification provides a more balanced interpretation of the findings and highlights areas for further optimization.

There is clear potential in this research, but I fear that the manuscript requires significant rewriting, improved articulation of the clinical context, and refinements in the presentation of technical content.

Reply:

            We sincerely appreciate the reviewer’s recognition of the potential impact of our research. In response to your concerns, we have undertaken extensive revisions throughout the manuscript to improve writing quality, clarify the clinical context, and enhance the presentation of technical content. These efforts included reorganizing key sections, refining language for clarity, and ensuring that both the clinical relevance and technical innovations are clearly articulated. We believe the revised manuscript now presents a clearer, more coherent, and impactful narrative.

Reviewer 2 Report

Comments and Suggestions for Authors

This study evaluates the Spectrum-Aided Vision Enhancer (SAVE) imaging modality as a tool for early esophageal cancer detection by applying several deep learning models (including YOLOv9, YOLOv10, YOLO-NAS, RT-DETR, and Roboflow 3.0) to hyperspectral-converted endoscopic images. The authors compare SAVE with conventional white-light imaging (WLI) and demonstrate that SAVE improves diagnostic metrics such as precision, recall, and F1-scores, particularly for detecting dysplasia and squamous cell carcinoma. Overall, the study suggests that incorporating SAVE into diagnostic workflows can enhance early-stage esophageal cancer detection under clinically relevant conditions.

The dataset details require clarification. The authors should specify how images were selected, the inclusion criteria, and the distribution of classes (normal, dysplasia, SCC, and bleeding) across the training, validation, and test sets.

Provide additional statistical metrics (e.g., confidence intervals for all model performance scores and p-values) to better compare the performance differences between SAVE and WLI.

Expand on the augmentation procedures to include more information on the types and frequency of augmentations applied and their impact on model generalization.

Explain any hyperparameter tuning processes in more detail to enhance reproducibility.

Clarify whether the analysis was performed on a static dataset or if any real-time detection capabilities were developed; if a real-time detection method was implemented, please report its performance and potential latency issues.

It is not clear from the manuscript whether the proposed framework includes a real-time detection component. If such a capability was developed, the authors should detail how the system performs in a clinical setting (e.g., response time and integration with clinical workflows). If not, a brief discussion on the potential for real-time application and future research directions would be beneficial.

The discussion should include a more thorough comparison with existing state-of-the-art methods. For instance, the authors could compare their performance with similar approaches that have employed hyperspectral imaging in esophageal cancer detection.

Citing additional studies on spectral imaging or similar deep learning diagnostic approaches would provide context for the significance of the improvements observed with SAVE.

The manuscript would benefit from a more streamlined and concise narrative, particularly in the methods and discussion sections, to avoid repetitiveness. Minor language edits and reorganization of paragraphs could enhance the overall readability and flow of the paper.

Author Response

Reviewer 2:

This study evaluates the Spectrum-Aided Vision Enhancer (SAVE) imaging modality as a tool for early esophageal cancer detection by applying several deep learning models (including YOLOv9, YOLOv10, YOLO-NAS, RT-DETR, and Roboflow 3.0) to hyperspectral-converted endoscopic images. The authors compare SAVE with conventional white-light imaging (WLI) and demonstrate that SAVE improves diagnostic metrics such as precision, recall, and F1-scores, particularly for detecting dysplasia and squamous cell carcinoma. Overall, the study suggests that incorporating SAVE into diagnostic workflows can enhance early-stage esophageal cancer detection under clinically relevant conditions.

Reply:

            Thank you for your accurate summary of our study and findings. We are pleased that the reviewer recognizes the relevance and potential clinical impact of incorporating SAVE with deep learning models to improve early-stage esophageal cancer detection. We appreciate your constructive feedback and will continue to refine the manuscript to ensure clarity and scientific rigor.

The dataset details require clarification. The authors should specify how images were selected, the inclusion criteria, and the distribution of classes (normal, dysplasia, SCC, and bleeding) across the training, validation, and test sets.

Reply:

            Thank you for your valuable comment. We have expanded the Materials and Methods section to clarify how images were selected and the inclusion criteria. Images were chosen based on confirmed histopathological diagnoses, high-quality visualization of the esophageal mucosa, and clear labeling into one of four classes: normal, dysplasia, squamous cell carcinoma (SCC), and inflammation. Poor-quality images or those lacking definitive diagnoses were excluded. We have also provided details on the dataset’s distribution, with images randomly split into training (70%), validation (20%), and test (10%) sets while maintaining balanced class proportions across these subsets.

Provide additional statistical metrics (e.g., confidence intervals for all model performance scores and p-values) to better compare the performance differences between SAVE and WLI.

Reply:

            Thank you for this valuable suggestion. We have now included 95% confidence intervals (CIs) for the precision, recall, and F1-score metrics for all models and lesion classes reported in Table 2. Additionally, we performed statistical testing comparing SAVE and WLI results, and p-values have been added to indicate the significance of the observed performance improvements. The enhancements in precision, recall, and F1-score with SAVE compared to WLI were statistically significant for most lesion classes, confirming the robustness of SAVE across different lesion types and models. These updates have been incorporated into the Results section and Table 2 to provide a clearer and more comprehensive comparison of model performances. We appreciate your feedback, which has strengthened the rigor and interpretability of our findings.

Expand on the augmentation procedures to include more information on the types and frequency of augmentations applied and their impact on model generalization.

Reply:

            Thank you for your suggestion. We have expanded the description of the data augmentation procedures in the Materials and Methods section. Additional details on the types of augmentations (horizontal flipping, random rotations, and shearing), their application frequency, and the rationale behind their selection have been included. We also clarified how these augmentations contributed to improving model generalization by introducing realistic variability and reducing overfitting. We have added the following:

“To enhance model generalization and mitigate overfitting, a comprehensive set of data augmentation techniques was applied to the training images. Each original image was augmented to create two additional versions, effectively tripling the training data volume. The augmentations included horizontal flipping (50% probability), random rotations (fixed at 90° and random rotations between −10° and +10°), and shearing transformations within a range of ±10°. These augmentations were applied uniformly across all lesion classes to maintain class balance. The frequency of each augmentation was randomized per image to ensure diversity in the augmented dataset. This approach aimed to simulate realistic variations in endoscopic imaging, such as different camera angles and tissue deformations, thereby improving the robustness of the deep learning models to variations encountered in clinical practice. By increasing the diversity of the training set, the augmentation strategy contributed significantly to the models’ ability to generalize to unseen data during validation and testing.”

            These revisions provide a clearer understanding of the augmentation strategy and its role in enhancing model performance.

Explain any hyperparameter tuning processes in more detail to enhance reproducibility.

Reply:

            Thank you for this important suggestion. We have now expanded the Materials and Methods section to provide a detailed explanation of the hyperparameter tuning process. This includes information on the specific hyperparameters adjusted, the search ranges used, the optimization strategies applied, and the criteria for selecting final values. We have also clarified that the tuning procedures were applied consistently across all models and both imaging modalities (WLI and SAVE) to ensure reproducibility and fair comparisons. We have specifically added, “The augmentation strategy enhanced the models' capacity to generalize to novel data during validation and testing by expanding the diversity of the training set. A system-atic hyperparameter tuning process was implemented for each deep learning model to enhance performance and ensure reproducibility. The main hyperparameters modified comprised the learning rate, batch size, weight decay, choice of optimizer, and detection confidence threshold. Each model was trained utilizing either Adam optimizer, with the selection corroborated by preliminary experiments assessing convergence rates and validation loss. The confidence threshold for object detection was adjusted between 0.4 and 0.7 to optimize precision and recall. Early stopping and learning rate scheduling were utilized to mitigate overfitting and guarantee stable convergence. The final hy-perparameters for each model were determined based on performance on the validation set, utilizing F1-score and mean Average Precision (mAP) as the principal evaluation metrics. All tuning procedures were uniformly executed across the WLI and SAVE da-tasets to guarantee an equitable comparison.”

            We appreciate your feedback, which has improved the transparency and reproducibility of our methodology.

Clarify whether the analysis was performed on a static dataset or if any real-time detection capabilities were developed; if a real-time detection method was implemented, please report its performance and potential latency issues.

Reply:

            Thank you for your comment. This study was conducted using a static dataset consisting of pre-collected and annotated endoscopic images. Real-time detection capabilities were not developed as part of the model evaluation process. However, to facilitate practical application, we developed a software application that allows users to convert white-light imaging (WLI) images into SAVE-enhanced images and subsequently apply the trained deep learning models for cancer detection. While this app supports streamlined image analysis, it operates in an offline, batch-processing mode rather than real-time video or live-stream detection. As such, latency measurements or real-time performance metrics were not applicable for this study. We have added the following in the discussion section.

It is not clear from the manuscript whether the proposed framework includes a real-time detection component. If such a capability was developed, the authors should detail how the system performs in a clinical setting (e.g., response time and integration with clinical workflows). If not, a brief discussion on the potential for real-time application and future research directions would be beneficial.

Reply:

            Thank you for this important observation. We confirm that the current framework was developed and tested using a static dataset, and real-time detection capabilities were not implemented as part of this study. However, we have developed a software application that converts WLI images into SAVE-enhanced images and applies the trained models for lesion detection in an offline setting. In response to the reviewer’s suggestion, we have now added a brief discussion in the manuscript highlighting the potential for future real-time application, including possible integration with endoscopic systems, anticipated latency considerations, and the need for further validation in clinical workflows. We have specifically added, “In subsequent endeavors, we intend to incorporate the SAVE imaging and deep learning detection models into a real-time clinical workflow. This will entail enhancing model inference speed, reducing latency, and verifying performance during live endoscopic procedures. Real-time deployment may substantially improve the early detection of esophageal cancer during routine endoscopy, thereby enhancing decision-making and patient outcomes.”

            We appreciate your feedback, which has allowed us to clarify the current scope of our work and outline future research directions.

The discussion should include a more thorough comparison with existing state-of-the-art methods. For instance, the authors could compare their performance with similar approaches that have employed hyperspectral imaging in esophageal cancer detection.

Reply:

            Thank you for this thoughtful suggestion. We would like to clarify that direct comparisons with existing hyperspectral imaging (HSI) methods are challenging because our approach represents a novel framework that converts standard WLI images into NBI-like hyperspectral-enhanced images using software-based techniques. In contrast, most previously reported HSI methods rely on specialized hardware systems, which differ fundamentally in both data acquisition and processing. As such, a direct performance comparison between hardware-dependent HSI studies and our software-driven method would not provide a meaningful or equitable evaluation. Nonetheless, we have emphasized the novelty and unique advantages of our approach in the revised Discussion and acknowledged the potential for future benchmarking as similar software-based methods emerge.

Citing additional studies on spectral imaging or similar deep learning diagnostic approaches would provide context for the significance of the improvements observed with SAVE.

Reply:

            Thank you for this valuable suggestion. In response, we have reviewed the recent literature and added citations to studies employing spectral imaging techniques and deep learning-based diagnostic approaches in gastrointestinal and esophageal cancer detection. These references provide context for the significance of our findings and highlight how SAVE advances current methods by offering a non-invasive, software-based alternative to hardware-dependent hyperspectral imaging systems. We appreciate your feedback, which has helped strengthen the scientific context and relevance of our study.

The manuscript would benefit from a more streamlined and concise narrative, particularly in the methods and discussion sections, to avoid repetitiveness. Minor language edits and reorganization of paragraphs could enhance the overall readability and flow of the paper.

Reply:

            Thank you for your constructive feedback. We have extensively revised and streamlined the manuscript, particularly the Methods and Discussion sections, to reduce repetitiveness and improve clarity. Additionally, minor language edits and paragraph reorganization have been implemented throughout the paper to enhance readability and ensure a more concise and coherent narrative. We appreciate your valuable suggestions, which have significantly improved the quality and flow of the manuscript.

Reviewer 3 Report

Comments and Suggestions for Authors
  1. Simple summary: This creates a model for detecting malignant tumors' stage and location. - Please rephrase.
  2. Abstract: Early detection is vital to improving - Please change to 'vital for'.
  3. Abstract: What does 'bleeding classes' mean in this context?
  4. Abstract is not structured, please change.
  5. Abstract's Results section is confusing, and it is not easy to interpret which information is relevant or precisely detailed.
  6. Reference list is not formatted according to requirements.
  7. Squamous cell carcinoma does not necessarily originate from squamous epithelium. It is defined as malignant epithelial tumour that shows squamous differentiation.
  8. Introduction Lines 67-77 contains inaccuracies. The diagnosis of EC is currently not based on imaging techniques, but upper GI endoscopy. The macroscopic signs of EC is unprofessional.
  9. Materials and Methods: Address of the hospital is not needed to be mentioned.
  10. This section should be written in past tense, and it should not contain results (for example, number of smokers is a result already).
  11. Use of betel nut is high for this group  - English language needs to be improved.
  12. Figure 1 legend:  Schematics of the whole study - Not professional
  13. Materials and methods: Instead of using a lot of mathematical formulas, proper, medical description should be provided about these tools.
  14. Results should start with the description of the examined population. 
  15. There are no references in the text for Figures or Tables.
  16. Figure 2: What does (line) mean?
  17. Table 2: Testing results - Please rephrase
  18. Figure 4. Detection results for WLI and SAVE - Please describe each, this is not proper description.
  19. Figure 5 is completely unnecessary.
  20. Some limitations of this study are worth discussing. - This is not how you start a discussion. It needs to be structured as: literature data and summary, your results, then comparing them to international data, then limitations, and advantages and take-home message.
  21. The discussion is not structured at all, and results are not summarized in it.
  22. The take-home-message is not clear at all.
Comments on the Quality of English Language

English language editing is necessary.
